# Fungal-Derived Mycoprotein and Health across the Lifespan: A Narrative Review

**DOI:** 10.3390/jof8070653

**Published:** 2022-06-22

**Authors:** Emma Derbyshire

**Affiliations:** Nutritional Insight, Epsom KT17 2AA, Surrey, UK; emma@nutritional-insight.co.uk

**Keywords:** ageing populations, fungi, glycaemic markers, human health, lifespan, lipids, mycoprotein

## Abstract

Mycoprotein is a filamentous fungal protein that was first identified in the 1960s. A growing number of publications have investigated inter-relationships between mycoprotein intakes and aspects of human health. A narrative review was undertaken focusing on evidence from randomized controlled trials, clinical trials, intervention, and observational studies. Fifteen key publications were identified and undertaken in early/young adulthood, adulthood (mid-life) or older/advanced age. Main findings showed that fungal mycoprotein could contribute to an array of health benefits across the lifespan including improved lipid profiles, glycaemic markers, dietary fibre intakes, satiety effects and muscle/myofibrillar protein synthesis. Continued research is needed which would be worthwhile at both ends of the lifespan spectrum and specific population sub-groups.

## 1. Introduction

Globally there has been a marked increase in longevity, however, double burdens of energy excess and undernutrition mean that physical deterioration, reduced life quality and increased medical costs are increasingly reported [1]. Projections of global human populations and the number of ‘peak people’ suggest that the population may plateau within the next 10 years [2]. Despite this, in the 1930s the global population was just a mere 2 billion which is anticipated to reach around 9.1 billion by 2050, with 2.1 billion expected to be aged 60 years or older [3,4,5]. The cumulative changes in population growth and transition towards ageing populations pose challenges to food systems, which need to meet the nutritional demands of expanding and ageing populations [6].

Shifts towards sustainable living mean that consumers are seeking to obtain food proteins that are non-animal derived yet aesthetically appealing with suitable nutritive value [7]. This includes ageing populations where sustainable protein sources could help to facilitate muscle mass and strength [6]. Fungal food proteins have been available for several decades, yet their health role(s) across the lifespan are only beginning to be fully appreciated [8]. Fungi are a predominant and diverse component of the Earths ecosystems [9]. Increasingly, fungi are being seen as a ‘Third Kingdom’ of organisms and valuable food protein outside the classical categories of animal and plant-derived foods [10,11,12].

There is some evidence that certain dietary patterns, such as veganism could have lifespan enhancing potential [13], however, for fungal-derived food proteins less is known about potential benefits across the lifespan. The present narrative review examines evidence looking at the potential roles of fungal mycoprotein consumption across the lifespan.

### Fungal-Derived Mycoprotein

Mycoprotein is a fungal-derived whole-food protein, filamentous fungal biomass and well-established meat analogue [14,15]. This fungal protein was first identified in the 1960s when it was derived from the soil-dwelling saprotrophic non-pathogenic micro fungus *Fusarium venenatum A3/5* [16,17]. Today mycoprotein is produced vertically in air lift pressure cycle fermenters 50 metres high, placing minimal demands on arable land compared with animal and plant-sourced proteins [14,18,19]. The use of microbial strains and specific substrates under various conditions (temperature, pH, relative humidity, moisture, inoculum age and size) has been found to yield mycoprotein biomasses produced using submerged fermentation with a high nutritional value [20]. Modelling has shown that, per unit of mass, cell-cultured foods such as mycoprotein have a lower environmental footprint that animal-derived proteins [21]. A recent paper using model projections found that substituting 20% of per-capita ruminant meat consumption with fermentation-derived microbial protein would (by 2050) reduce annual deforestation and related carbon dioxide emissions by about half, reduce methane emissions and offset expansions in global pasture areas [22].

From a nutritional stance mycoprotein provides a range of nutrients of value across the lifespan. It provides the nine main essential amino acids and has a protein digestibility-corrected amino acid score of 0.996 indicating that it is a high-quality protein [11,23]. According to European Commission standards mycoprotein can be classified as being ‘high in fibre ‘due to it providing at least 6g of fibre per 100 g [24,25]. It is also low in both total and saturated fat and contains negligible amounts of cholesterol [26]. Mycoprotein provides a range of micronutrients including vitamin B12, riboflavin, folate, phosphorous, choline, zinc and manganese [27].

## 2. Materials and Methods

A narrative review was undertaken using the National Library of Medicine, National Center for Biotechnology Information (PubMed.gov (accessed on 12 May 2022)) database with the predefined keyword ‘mycoprotein’.

The literature search was finalized on 9th June 2022. The selection criteria were limited to randomized controlled trials, clinical trials, intervention studies and observational studies. The search was limited to humans, and those published in English language. Fifteen key publications were retrieved and are discussed in the present narrative review (Table 1). An additional search for publications was conducted by reviewing reference lists. Abstract or poster presentations were excluded.

## 3. Fungal Mycoprotein across the Lifespan

### 3.1. Early and Young Adulthood

Early adulthood has been defined as ages 20 to 24 years and young adulthood ages 26 to 31 years [43,44]. Certain lifestyle factors, such as preventing weight gain in early adulthood may be important in preventing premature death later in life [45]. Additionally, other work has shown that higher intakes of dietary fibre during early adulthood in females was associated with reduced breast cancer risk [46]. Like adolescence, early adulthood is a life stage where opportunities for health are vast [47].

Eight studies have evaluated the effects of mycoprotein consumption in relation to markers of health in early and young adulthood [28,29,30,31,32,33,34,35]. A range of outcomes were studied including effects of ingestion on the plasma lipidome [28], cholesterol levels [35] protein synthesis rates [31,32], postprandial amino acid, glucose and insulin levels [29,30,33] and energy intake and appetite [34].

Coelho et al. (2021) undertook a 7-day fully controlled diet where lunch and dinner contained either mycoprotein or meat/fish as the control group [28]. After 1-week mycoprotein ingestion by adults aged 24 ± 1 year significantly reduced total plasma cholesterol, free cholesterol, low-density lipoprotein cholesterol and high-density lipoprotein-2 cholesterol [28]. Another randomized trial by the same team conducted with ten healthy young adults found that nucleotide-rich mycoprotein ingestion (as a mixed-meal) did not influence postprandial blood glucose nor serum insulin concentrations [29].

Monteyne et al. (2020) allocated participants (22 ± 1 year) to receive ingest either 70 g mycoprotein or 31 g milk protein following a bout of resistance exercise [31]. Results showed that the myoprotein consumption enhanced postexercise and resting muscle protein synthesis rates to a greater extent than the control [31]. Similarly, work by the same research group conducted with young males (mean age 22 years) found that a 35 g dose of branched chain amino acid (BCAA) enriched mycoprotein stimulated muscle protein synthesis rates at rest and postexercise but to a lesser extent than the 70 g mycoprotein bolus (BCAA matched) in healthy young men [32].

Earlier research by Dunlop and colleagues (2017) provided healthy young males (mean age 28 years) a test drink containing either a bolus of mycoprotein (20, 40, 60 or 80 g) or 20 g of milk protein which were mass matched [30]. Mycoprotein resulted in slower and more sustained hyperinsulinaemia and hyperaminoacidaemia compared with milk, indicating that this dietary protein could have the potential to facilitate rates of muscle protein synthesis [30]. Turnbull & Ward (1995) recruited young adults (mean age 22.8 years) providing them with milkshakes with or without mycoprotein and finding that glycemia and insulinemia were significantly reduced at 60 and 30 min respectively after ingestion [33]. In other earlier work Turnbull et al. (1993) recruited thirteen females aged 24.8 ± 7.9 years finding that desire to eat and levels of food consumption reduced after myoprotein ingestion, compared with chicken as a control [34].

### 3.2. Adulthood (Mid-Life)

Mid-life is often referred to as ‘the halfway mark’ [48]. Health outcomes later in life are often determined and underpinned by activities that occurred in midlife [49]. A growing body of evidence has linked certain dietary factors, including lower protein intakes during adulthood to increased risk of muscle loss (sarcopenia), functional limitations and increased the risk of falls [50,51]. Interestingly, other research has suggested that changes in gut microbiota i.e., *Bifidobacterium* reduction begin to change as early as midlife [52]. At least six studies have investigated the effects of mycoprotein ingestion of markers of health during adulthood [36,37,38,39,40,41].

Cross-sectional research analysing data from free-living adults taking part in the United Kingdom National Diet and Nutrition Survey showed that mycoprotein consumers had significantly higher intakes of dietary fibre, improved diet quality and lower markers of glycaemia [37].

Bottin et al. (2016) undertook a randomized controlled trial with overweight and obese adults (18 to 65 years) finding that mycoprotein ingestion (44, 88 or 132 g) reduced insulin levels and energy intake compared with a chicken control [36]. Williamson and researchers (2006) found that a mycoprotein and tofu preload had satiating effects amongst 41 pre-menopausal overweight females, compared with a chicken preload [41]. A 6-week intervention trial by Ruxton & McMillan (2010) found that mycoprotein consumption (88g daily, wet weight) led to significant reductions in total and low-density lipoprotein cholesterol, particularly amongst those with high levels at baseline [38]. Larger and longer blinded trials in community settings would build on these results.

Earlier research undertaken by Turnbull et al. (1992) recruited adults aged 25 to 61 years finding that mycoprotein ingestion (26.9 g dry weight/day over 8-weeks reduced total and low-density lipoprotein cholesterol compared with control biscuits [39]. In a shorter 3-week study adults 19–48 years received 191 g mycoprotein daily distributed over lunch and dinner, with a 13% reduction in plasma cholesterol observed compared with the meat control [40].

### 3.3. Older/Advanced Age

A range of definitions have been used to encompass older age. The term ‘old-old’ has been used to define those aged 80 years and over [53]. Other publications use the term ‘advanced age’ to study those typically over 65 years of age [54]. Age, however, is highly subjective, particularly at this life stage. For example, in one study perceived mean age of ‘old age’ was around 69 years, with some participants reporting feeling younger than their chronological age [55].

The World Health Organization forecasts that the proportion of people aged 85 years and over in the European region is predicted to increase from 14 million to 19 million by 2020, and double further to 40 million by 2050 [56]. On a global scale an estimated 2 billion people are expected to be older than 65 years by 2050, potential having heavy impacts on health and social care sectors if frailty is not addressed [57]. It is well established that good nutrition can support healthy ageing, helping to get ahead of age-related declines in physiology [58].

Research has now investigated the effects of mycoprotein ingestion in older age. A randomised controlled trial recruited older adults (n = 19; 66 ± 1 year) allocating these to consume a 3-day isocaloric high-protein diet deriving protein from mycoprotein or animal protein [42]. The team found that the fungal (vegan-derived) mycoprotein could facilitate rested and exercised daily myofibrillar protein synthesis rates similarly to that amongst those on the omnivorous diet [42].

## 4. Discussion

Genetic, environmental and lifestyle factors which include nutrition form an integral part of our health [59]. Multimorbidity defined as the coexistence of two or more health conditions is increasingly being linked to considerable economic burdens [60]. This is emerging to be one of the greatest challenges to health services, partially driven by ageing populations [61]. It has been estimated that by 2035 17% of the population in the United Kingdom alone is expected to have four or more chronic conditions, with functional decline often tending to form part of this [61,62].

From a dietary stance lifespan health can be influenced by certain nutrients [63]. For example, it has been proposed that shifts away from animal-derived food proteins providing high levels of methionine and movements towards alternative plant sources may benefit longevity and metabolic health [63]. There is also a plethora of evidence linking dietary fibre intake to metabolic health, colonic health and reduced cardiovascular and mortality risk [64]. Mycoprotein has been positioned as a fungal protein that is high-fibre, protein rich and produced sustainably [65]. Given this, it has been questioned whether fungal-derived food protein such as mycoprotein should be included more prominently within food-based dietary guidelines [11,12].

A growing body of evidence from systemic reviews shows that acute mycoprotein ingestion reduces energy intake and insulinaemia [33,37,66] and has the ability to lower circulating cholesterol levels, improve postprandial glycaemic control and facilitate satiety [33,35,38,65,67]. Evidence from the present narrative review further demonstrates that fungal mycoprotein could benefit health across the lifespan.

In early and young adulthood benefits have been observed on the plasma lipidome [28], in sustaining insulin and amino acids levels [30], attenuating the desire to eat (inducing satiety) [34] and stimulating resting and postexercise muscle protein synthesis rates [32]. In adulthood most research has been conducted on healthy adult populations. Observational evidence shows that mycoprotein consumers have higher dietary quality scores, fibre intakes and lower glycaemic markers and a lower body mass index [37]. Trials conducted on overweight adults show that mycoprotein ingestion was associated with reduced energy intake and insulin release [36] and satiety effects [41]. Several other publications found that mycoprotein ingestion improved blood lipid levels [35,38,39,40]. Research with older adults (mean age 68 years) showed that induced myofibrillar protein synthesis rates aligned with those followed an omnivorous diet [42]. Continued research is now needed across additional population groups. Larger studies would be worthwhile as trial sample sized ranged from ten to 100. This could have contributed to discrepancies between study findings.

Overall, these are interesting findings implying that fungal-derived mycoprotein has an important role to play in health across the lifespan. Regarding, mechanistic effects the ingredient structure of mycoprotein is what appears to lower lipolysis and bind bile salts which, in turn, is thought to lower blood lipid levels [68]. Regarding the high bioacessibility of protein from fungal mycoprotein, this is thought to be attributed to porous cell walls which facilitate the diffusion of proteases [69].

A publication modelling fibre intake in childhood showed one daily portion of mycoprotein shapes for children aged 4-to-10 years and 11-to-18 years could contribute to approximately a quarter of the daily fibre recommendation [70]. It would be useful to conduct additional randomized controlled trials in childhood and ageing populations. There is also scope to investigate inter-relationships between fungal mycoprotein ingestion and markers of health at specific lifespan phases such as gestation. Great opportunity lies in accruing evidence for non-traditional dietary proteins such a mycoprotein given rising demands for health sustainably-produced protein foods [71].

## 5. Conclusions

This narrative review has described the roles of fungal mycoprotein across the lifespan. Given expanding and ageing populations coupled with growing multimorbidity’s the integration of fungal mycoprotein with daily diets could help to diversify protein intakes and have potential beneficial implications for health across the lifespan. Further research at both ends of the lifespan spectrum would be worthwhile.

## Figures and Tables

**Table 1 jof-08-00653-t001:** Mycoprotein & Health Across the Lifespan: Key Studies.

Life Stage (Author, Year, Location)	Population (Sample Size, Age, Health)	Study Design	Methods	Health Outcome(s)	Main Findings
*Early and young adulthood*					
Coelho et al. (2021) UK [28]	*n* = 20, 24 years, recreationally active.	Randomised, parallel-group trial.	A 7-d fully controlled diet where lunch & dinner contained either meat/fish or mycoprotein as the source of dietary protein.	Plasma lipidome.	Substituting meat/fish for mycoprotein twice daily for 1 week resulted in a reduction in cholesterol-containing lipoproteins.
Coelho et al. (2020) UK [29]	*n* = 10, 25 years	Randomized, controlled, double-blind, crossover trial.	Consumed a mixed-meal containing nucleotide-depleted mycoprotein or high-nucleotide mycoprotein on two separate visits.	Postprandial glucose, Serum insulin, Serum uric acid	The nucleotide-rich mixed-meal increased serum uric acid concentrations for ~12 h, but had no effects on postprandial blood glucose or serum insulin levels.
Dunlop et al. (2017) UK [30]	*n* = 12, 28 years, healthy young males.	Randomised, single-blind, cross-over study.	Volunteers consumed a test drink containing either 20 g milk protein or a bolus of mycoprotein (20 g, 40 g, 60 g or 80 g).	Postprandial hyperaminoacidaemia, hyperinsulinaemia	Mycoprotein ingestion resulted in slower but more sustained hyperinsulinaemia & hyperaminoacidaemia compared with milk when protein matched.
Monteyne et al. (2020a) UK [31]	*n* = 10, 22 years, healthy young males.	Randomized, double-blind, parallel-group study.	Participants received infusions of L-phenylalanine and ingested either 31 g milk protein or 70 g mycoprotein following a bout of unilateral resistance-type exercise	Protein synthesis rates	Mycoprotein ingestion stimulated resting & postexercise MPS rates to a greater extent than a leucine-matched bolus of milk protein.
Monteyne et al. (2020b) UK [32]	*n* = 10, 22 years, young males	Randomized, double-blind, parallel-group study.	Participants received infusions of L-phenylalanine ingested with either 70 g mycoprotein or 35 g BCAA-enriched mycoprotein following a bout of unilateral resistance exercise.	Protein synthesis rates	The lower-dose BCAA-enriched mycoprotein stimulated resting and postexercise MPS rates, but to a lesser extent compared with the ingestion of a BCAA-matched 70-g mycoprotein bolus.
Turnbull & Ward (1995) UK [33]	*n* = 19, 22.8 ± 3.55 years	Single meal study periods x2, crossover design	Milkshakes provided containing mycoprotein or a control.	Glycemia, insulinemia	Glycemia was significantly reduced 60 min after mycoprotein ingestion. Insulinemia was significantly reduced 30 min (19% reduction) and 60 min (36% reduction) after mycoprotein ingestion.
Turnbull et al. (1993) UK [34]	*n* = 13 24.8 ± 7.8 years, female.	3-day study periods x2	Subjects ingested an isoenergetic meal providing mycoprotein or chicken.	Energy intake, appetite	Food consumption and desire to eat decreased after mycoprotein compared with chicken consumption.
Udall et al. (1984) USA [35]	*n* = 100, 19.9–25.6 years	30-day double-blind cross-over study	Cookies with or without 20 g of *Fusarium graminearium* were ingested.	Tolerance, cholesterol levels	There was a decrease in serum cholesterol during the *F graminearium* study.
*Adulthood (mid-life)*					
Bottin et al. (2016) UK [36]	*n* = 55, 18–65 years, overweight & obese adults.	Two randomised-controlled trials.	Consumed a test meal containing low (44 g), medium (88 g) or high (132 g) mycoprotein or isoenergetic chicken.	Postprandial insulin release	Mycoprotein ingestion reduced energy intake & insulin release in overweight volunteers.
Cherta-Murillo et al. (2021) UK [37]	*n* = 5507 free-living adults	Observational data used from the UK NDNS years 2008/9 to 2016/17.	Cross-sectional secondary analysis of the UK NDNS years 2008/9 to 2016/17.	Fibre intake, energy density intake, BMI, fasting glucose, glycated HbA1c	Mycoprotein consumers had higher dietary fibre intakes, lower glycaemic markers, energy density intake & BMI than non-consumers.
Ruxton & McMillan (2010) UK [38]	*n* = 21, 17–58 years	6-week non-blinded, controlled intervention	Asked to eat mycoprotein (88g wet weight mycoprotein), daily for six weeks.	Total cholesterol levels, glucose levels	Good compliance with the mycoprotein-rich diet appeared to significantly lower total blood & LDL cholesterol.
Turnbull et al. (1992) UK [39]	*n* = 21, aged 25–61 years staff & students.	8-week study.	The experimental group was fed cookies containing mycoprotein and the control group a cookie without mycoprotein	Blood lipids	Total cholesterol was reduced by 0.95 mmol/L in the mycoprotein versus 0.46 mmol/L in the control group. LDL was reduced by 0.84 mmol/L in the mycoprotein group versus 0.34 mmol/L in the control group.
Turnbull et al. (1990) UK [40]	*n* = 17, 19–48 years, staff & students.	3-week study	The experimental group was fed mycoprotein instead of meat and the control diet included meat.	Blood lipids	LDL declined in the mycoprotein group by 9%.
Williamson et al. (2006) USA [41]	*n* = 42, 18–50 years, pre-menopausal overweight females.	3-test day interventions.	At lunch, isocaloric pasta preloads, containing mycoprotein, tofu, or chicken provided.	Eating behaviour, hunger, satiety	Mycoprotein & tofu versus the chicken preload were associated with lower food intake after the preload at lunch indicating satiating properties.
*Older/Advanced age*					
Monteyne et al. (20021) UK [42]	*n* = 10, 68 ± 2 years, healthy older adults.	Randomized, parallel-group, controlled trial.	3-day isocaloric high-protein (1.8 g·kg body mass^−1^·d^−1^) diet, where the protein was from mycoprotein providing 57% of daily protein intake.	Myofibrillar protein synthesis rates	Mycoprotein & omnivorous protein sources supported rested & exercised daily myofibrillar protein synthesis rates in healthy older adults ingesting a high-protein diet.

Key: BCAA, branched chain amino acids; BMI, Body Mass Index; HbA1c, glycated hemoglobin; LDL, low-density lipoprotein; MPS, muscle protein synthesis; NDNS, National Diet & Nutrition Survey; UK, United Kingdom.

## Data Availability

Not applicable.

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
