# Peer review of "Fungal-Derived Mycoprotein and Health across the Lifespan: A Narrative Review"

_jof, 2022, doi:10.3390/jof8070653_

Round 1

Reviewer 1 Report

The  author made a narrow review in thhe effect of the fungal mycoprotein on human health. The manuscript is well organized and well-writen, the main findings of this manuscript is useful and deserved publishing. However, the main weak for this manuscript is there are only 11 papers was involved and this made the conclusion of this manusccript is unreliable. I suggest the author to try to get more references in this study.

Author Response

Thank you for the feedback. The search criteria has been broadened a touch and additional papers added as per feedback. Thank you. 

Reviewer 2 Report

Mycoprotein intakes and aspects of human health have closely interrelationships. This narrative review described the roles of fungal mycoprotein across the lifespan mainly focusing on evidence from clinical trials, randomized controlled trials and observational studies. And the findings showed that fungal mycoprotein could contribute to an array of health benefits across the lifespan including improved lipid profiles, glycaemic markers, dietary fibre intakes, satiety effects and muscle/myofibrillar protein synthesis. After reading the manuscript carefully, I can tell you that the information it brings to readers is reasonable and interesting. The work will make a major contribution to the field of fungal cell biology, metabolism and physiology. In my opinion, only a few minor versions, such as English, grammar and typos should be revised, also the format of references needs to be unified, et al..

Author Response

Thank you for the valuable feedback. The publication has bee re-read thoroughly and re-edited. As per the other reviewers feedback the search criteria has been broadened a touch to enable a few additional papers to be added. The conclusions are thus reinforced further. Thank you for your your comments.

Round 2

Reviewer 1 Report

Thanks for the author and I am glad more references were added in this manuscript tp reinforce the results. All the issues I concerned were solved and I recommended it for publishing.